# Tracking transcription factor mobility and interaction in Arabidopsis roots with fluorescence correlation spectroscopy

**Natalie M Clark[1,2], Elizabeth Hinde[3], Cara M Winter[4], Adam P Fisher[1], Giuseppe Crosti[4], Ikram Blilou[5], Enrico Gratton[3], Philip N Benfey[4]\*, Rosangela Sozzani[1]\***

[1]Department of Plant and Microbial Biology, North Carolina State University, Raleigh, United States; [2]Biomathematics Graduate Program, North Carolina State University, Raleigh, United States; [3]Laboratory for Fluorescence Dynamics, University of California, Irvine, Irvine, United States; [4]Department of Biology, Howard Hughes Medical Institute, Duke University, Durham, United States; [5]Plant Developmental Biology, Wageningen University, Wageningen, Netherlands

**Abstract** To understand complex regulatory processes in multicellular organisms, it is critical to be able to quantitatively analyze protein movement and protein-protein interactions in time and space. During Arabidopsis development, the intercellular movement of SHORTROOT (SHR) and subsequent interaction with its downstream target SCARECROW (SCR) control root patterning and cell fate specification. However, quantitative information about the spatio-temporal dynamics of SHR movement and SHR-SCR interaction is currently unavailable. Here, we quantify parameters including SHR mobility, oligomeric state, and association with SCR using a combination of Fluorescent Correlation Spectroscopy (FCS) techniques. We then incorporate these parameters into a mathematical model of SHR and SCR, which shows that SHR reaches a steady state in minutes, while SCR and the SHR-SCR complex reach a steady-state between 18 and 24 hr. Our model reveals the timing of SHR and SCR dynamics and allows us to understand how protein movement and protein-protein stoichiometry contribute to development.

**\*For correspondence:** philip. benfey@duke.edu (PNB); ross_sozzani@ncsu.edu (RS)

**Competing interests:** The authors declare that no competing interests exist.

## Introduction

During development, multicellular organisms must coordinate patterning, maintenance, and growth of different cell types. This dynamic coordination is achieved through complex spatio-temporal signaling mechanisms (*Voas and Rebay, 2004*; *Scheres, 2007*; *Han et al., 2014*; *Sozzani and Iyer-Pascuzzi, 2014*; *Sablowski, 2015*; *Tuazon and Mullins, 2015*; *Fisher and Sozzani, 2016*). Many of the signaling mechanisms regulating development utilize mobile signals that cross cell boundaries (*Koizumi et al., 2011*; *Xu et al., 2011*; *Gallagher et al., 2004*). In plants, the intercellular movement of transcription factors and the spatio-temporal control of protein complex formation regulate many processes including cell fate specification.

The key to a systems-level understanding of development in multicellular organisms is the ability to obtain quantitative information about various signaling factors. In the *Arabidopsis* root, the SHORTROOT-SCARECROW (SHR-SCR) mediated gene regulatory network (GRN) is a well-characterized developmental pathway that controls ground tissue patterning and endodermal cell fate specification (*Levesque et al., 2006*; *Cui et al., 2007*; *Sozzani et al., 2010*; *Cruz-Ramırez et al., 2012*; *Long et al., 2015*; *Moreno-Risueno et al., 2015*). SHR is transcribed in the vasculature (*Helariutta et al., 2000*; *Nakajima et al., 2001*), and then the protein moves to the adjacent cell

**eLife digest** Stem cells are a specific type of cell found in both plants and animals. These cells can divide to produce daughter cells that can take on the role of any of the different tissues and organs within the plant or animal. A plant known as *Arabidopsis* is often used as a model in scientific research. In *Arabidopsis*, two proteins called SHORTROOT and SCARECROW are known to control the ability of stem cells in the roots to divide.

SHORTROOT is made in cells at the center of the root known as the vasculature. From there, it moves to the next cell layer (called the endodermis) where it interacts with SCARECROW to form a protein complex. Here, Clark et al. investigated how quickly SHORTROOT moves between cells, the direction it moves in, and how it interacts with SCARECROW.

The experiments used a new imaging technique called scanning fluorescence correlation spectroscopy to track the movements of SHORTROOT molecules in the root. This technique relies on the protein of interest (in this case, SHORTROOT) being attached to a fluorescent protein so that it is visible when the cells are examined. In plants that had lower levels of SCARECROW, SHORTROOT moves between cells more quickly and in an unrestricted manner. This suggests that SCARECROW forms a complex with SHORTROOT to restrict its movement in the endodermis. The experiments also show that SHORTROOT is only able to leave the endodermis to return to the vasculature when SCARECROW levels are lower than normal.

Clark et al. developed a model to map the behavior of SHORTROOT and SCARECROW in the root and predict how the levels of these proteins change over time. One of the next steps following on from this work would be to test whether other proteins restrict the movement of SHORTROOT, perhaps by studying mutant plants in which SHORTROOT is less able to move.

layer where it is retained in the nuclei of the Quiescent Center (QC), Cortex/Endodermis Initials (CEI) and endodermis (*Nakajima et al., 2001*; *Gallagher et al., 2004*; *Gallagher and Benfey, 2009*). In these cells, SHR activates the expression of the downstream transcription factor SCR (*Helariutta et al., 2000*; *Levesque et al., 2006*; *Cui et al., 2007*; *Sozzani et al., 2010*), which, as shown by yeast two hybrid experiments, interacts with SHR and prevents further SHR movement (*Heidstra et al., 2004*; *Cui et al., 2007*; *Long et al., 2015*). Although there have been important advances in identifying the essential features that govern the SHR-SCR GRN (*Gallagher et al., 2004*; *Sena et al., 2004*; *Cui et al., 2007*; *Gallagher and Benfey, 2009*; *Sozzani et al., 2010*; *Cruz-Ramırez et al., 2012*), the ability to measure key network parameters that may contribute to patterning and cell fate specification remains a fundamental bottleneck.

New imaging tools that enable parameter quantification and acquisition of in vivo kinetic parameters could provide quantitative information that describes temporal and spatial dynamics of proteins in multicellular organisms. Thus, we explored the possibility of combining scanning Fluorescence Correlation Spectroscopy (scanning FCS) techniques. Unlike more common time correlation FCS techniques, which only use temporal information, scanning FCS techniques utilizes both the spatial and temporal information present in a confocal raster scan to measure protein movement, protein-protein interactions, and the stoichiometry of protein complexes. (*Petrasek and Schwille, 2008*; *Digman and Gratton, 2011*). Previously, these techniques have only been used to quantify protein mobility and the dynamics of protein association in cell cultures (*Digman et al., 2005a*; *2005b*; *Digman and Gratton, 2009a*; *Digman et al., 2009a*; *Jameson et al., 2009*; *Rossow et al., 2010*; *Hinde et al., 2010*; *2011*; *Vetri et al., 2011*). We combined the techniques of Raster Image Correlation Spectroscopy (RICS), Pair Correlation Function (pCF) and Number and Brightness (N&B) to analyze SHR and SCR mobility and interaction at high spatio-temporal resolution. By using RICS and pCF (*Digman et al., 2005a*; *2005b*; *Brown et al., 2008*; *Digman and Gratton, 2009a*; *2009b*; *Jameson et al., 2009*; *Rossow et al., 2010*; *Hinde et al., 2010*; *2011*; *Vetri et al., 2011*; *Digman and Gratton, 2011*), we quantified the rate and directionality of SHR movement. Specifically, we used RICS and a 3D Gaussian diffusion model (*Digman et al., 2005a*; *2005b*) to calculate the diffusion coefficient of SHR in different root cell types. We also acquired line scans and performed pCF analyses (*Hinde et al., 2010*) to assess the directionality of SHR movement in these different cell types. Moreover, we used N&B and cross-

correlation analyses (*Digman et al., 2008*; *2009a*; *2009b*) to characterize the oligomeric state of SHR and the stoichiometry of the SHR-SCR complex, respectively. Finally, we incorporated the diffusion coefficient of SHR and the stoichiometry of the SHR-SCR complex into a mathematical model of SHR and SCR dynamics. Our results demonstrate that these parameters can be used in predictive mathematical models, allowing us to understand how protein movement and stoichiometry of protein complexes contribute to developmental processes. Further, our study highlights how these non-invasive scanning FCS techniques can be used to experimentally measure protein movement and protein-protein interactions within multicellular organisms.

## Results

### Protein movement quantified using raster image correlation spectroscopy (RICS)

A key parameter in biological models is molecular diffusion, which is frequently estimated based on published measurements from single cell organisms (*Spiller et al., 2010*). In order to measure protein movement in a multicellular organism such as *Arabidopsis,* we used Raster Image Correlation Spectroscopy (RICS), which returns an autocorrelation function (ACF) by correlating fluorescence intensity fluctuations in pixels in an image over time and space. The diffusion coefficient (DC) is then calculated by fitting the ACF with a diffusion model (*Figure 1*). Since the fit of the diffusion model depends on the choice of imaging parameters, such as pixel size and pixel dwell time, we first set these parameters by performing RICS analysis on Green Fluorescent Protein (GFP) driven by the CaMV 35S constitutive promoter (35S:GFP) (*Table 1*, see Materials and methods). In the *Arabidopsis* root, the resulting DC for free GFP was $6.33 \pm 0.37$ $\mu m^2$/s ($n = 34$ for different cells, including vascular, endodermal, and QC cells) (*Figure 1—figure supplement 1*). We obtained similar diffusion coefficients using two different confocal microscopes (Zeiss 780 and Zeiss 710).

We next used RICS to determine if the DC of SHR differs between the vasculature, where it is produced, and the endodermis and QC, where it moves and interacts with SCR (*Heidstra et al., 2004*; *Cui et al., 2007*; *Long et al., 2015*). To this end, we used a SHR:SHR-GFP translational fusion, which complements the *shr2* mutant phenotype (SHR:SHR-GFP in *shr2*) (*Nakajima et al., 2001*). Our RICS analysis showed that SHR moves at a rate of $2.45 \pm 0.26$ $\mu m^2$/s ($n = 40$) and $1.73 \pm 0.15$ $\mu m^2$/s ($n = 16$) within a population of vascular and endodermal cells and vascular and QC cells, respectively (*Figure 1*). Notably, in a population of only endodermal cells, SHR moves at a significantly slower rate ($1.29 \pm 0.14$ $\mu m^2$/s $n = 19$, Wilcoxon with Steel-Dwass, $p = 0.0303$) (*Figure 1*). To investigate whether the reduction in SHR movement in the endodermis may be due to its interaction SCR (*Cui et al., 2007*; *Levesque et al., 2006*; *Sozzani et al., 2010*; *Moreno-Risueno et al., 2015*), we measured the DC of SHR in a SCR RNAi (SCRi) line (SHR:SHR-GFP in SCRi) (*Cui et al., 2007*). In this line, in which the levels of SCR are reduced, SHR diffusion in populations of only endodermal cells was similar to that in populations of vascular and endodermal cells ($2.40 \pm 0.29$ $\mu m^2$/s, $n = 14$, Wilcoxon with Steel-Dwass, $p = 0.9337$) (*Figure 1*). Overall, our RICS data are in agreement with genetic and molecular data that show SHR movement is restricted by SCR (*Heidstra et al., 2004*; *Cui et al., 2007*; *Long et al., 2015*). Most importantly, our RICS data provide a precise quantification of how SHR diffusion is affected by the presence of SCR.

### The direction of protein movement determined using pair correlation function (pCF) analysis

An open question is whether SHR moves unidirectionally, only from the inner to the outer cell layers, or in a bidirectional fashion. The former would be consistent with an active transport mechanism from the vasculature into the endodermis. To detect the route of intercellular movement of SHR between the vasculature and endodermis as well as between the endodermis and cortex, we used pair correlation function (pCF) analysis on line scans (*Figure 2*). We used pCF to measure the directionality of movement by correlating pixels that are separated by a specific pixel distance (*Hinde and Cardarelli, 2011*). To account for differences in cell size, as well as cell wall orientation within our images, we used three different distances: 5 pixels, 7 pixels, and 9 pixels. Generally, the pCF analysis returns a carpet, or heatmap, that shows the fluorescence correlation over time (y-axis) and space (x-axis). If proteins move across the cell wall, there is an arch that represents the delay in movement. If proteins are unable

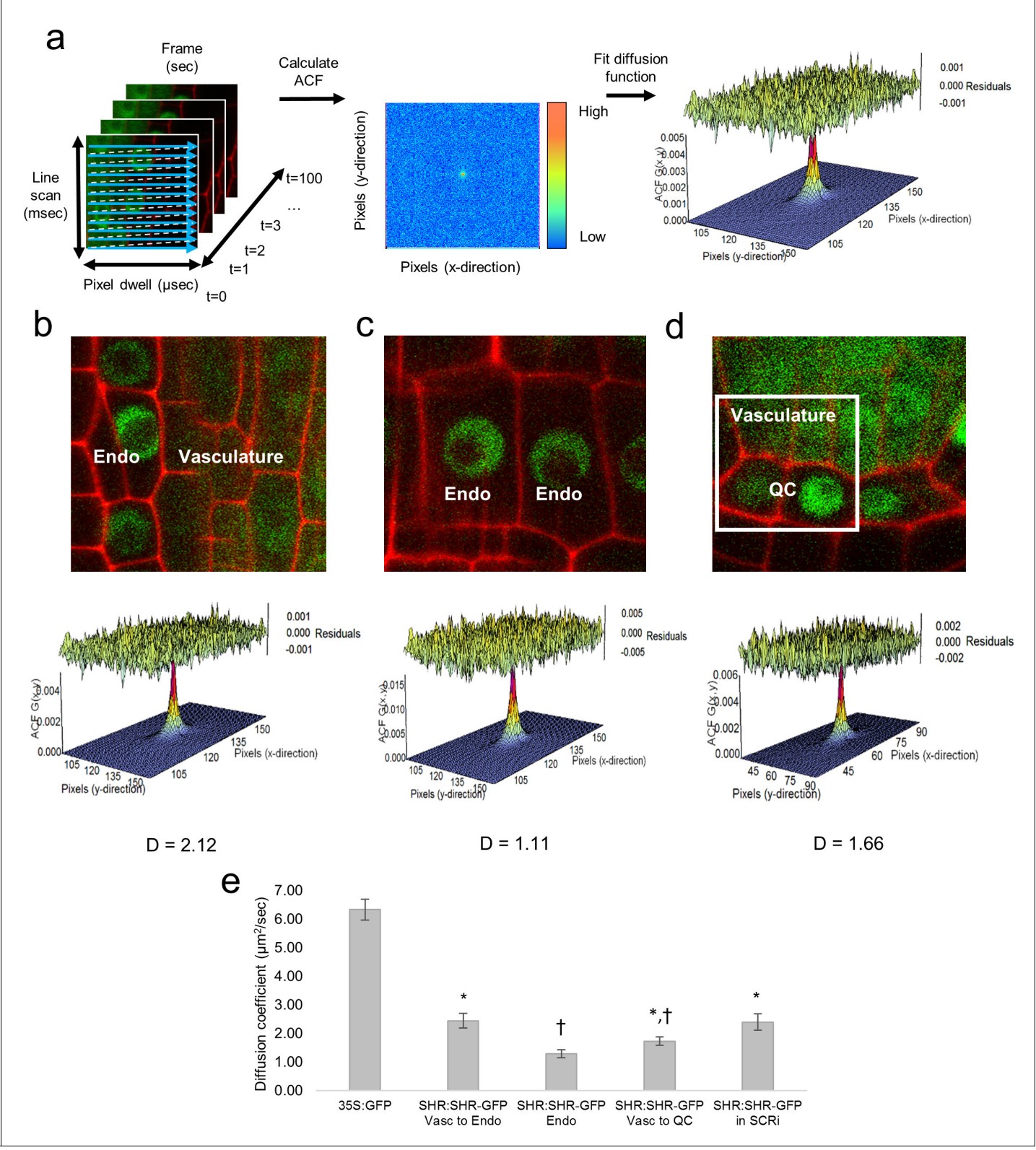

**Figure 1.** Diffusion coefficients obtained by performing RICS on SHR:SHR-GFP in *shr2*. (a) Schematic showing image acquisition and RICS analysis. (Left) A time series of 100 frames (time points) acquired using predetermined imaging parameters (*Table 1*). (Middle) Autocorrelation function (ACF) calculated from the time series. Red represents a high ACF value, blue represents a low ACF value. (Right) Fit of the ACF to a Gaussian diffusion model to calculate the diffusion coefficient. (b–d) Representative images of SHR:SHR-GFP in *shr2* taken in regions containing the vasculature and endodermis

*Figure 1 continued on next page*

*Figure 1 continued*

(**b**), endodermis only (**c**), vasculature and QC (**d**). Cell walls are marked in red using propidium iodide (PI). Below each image is its ACF fit using the Gaussian model and the calculated diffusion coefficient for that representative image. (**d**) 128x128 pixel region of interest (ROI) used for RICS (white frame). (**e**) Bar graph showing average diffusion coefficients of 35S:GFP (n = 34), SHR:SHR-GFP in *shr2* (n = 40) for vasculature and endodermis, n = 19 for endodermis, n = 20 for vasculature and QC) and SHR:SHR-GFP in SCRi (vasculature and endodermis, n = 14). Groups that have different symbols are significantly different from each other and from the 35S:GFP line (Wilcoxon with Steel-Dwass, p<0.05). Error bars are s.e.m. Source data is provided in *Figure 1—source data 1–4* .

The following source data and figure supplement are available for figure 1:

**Source data 1.** Diffusion coefficient of 35S:GFP line obtained using RICS with the Zeiss 780 and Zeiss 710 instruments.

**Source data 2.** Diffusion coefficient of SHR:SHR–GFP in *shr2* line obtained using RICS with the Zeiss 780 and Zeiss 710 instruments.

**Source data 3.** Diffusion coefficient of SHR:SHR–GFP in SCRi line obtained using RICS with the Zeiss 780 instrument.

**Source data 4.** Statistical analysis of diffusion coefficients obtained by RICS.

**Figure supplement 1.** RICS analysis on the 35S:GFP line.

to cross the barrier, the arch is absent (*Hinde et al., 2010*) (*Figure 2*) (see Materials and methods). Thus, we performed a binary analysis on each carpet to determine the movement of SHR between cells by looking at the presence, or absence, of these arches. Specifically, we recorded a 1 if the carpet showed an arch and a 0 if no arch was present (*Figure 2*). We took the average of these values from the different pixel distances (5, 7, and 9) to represent one biological replicate. We then calculated the protein Movement Index (MI), which is the average of all biological replicates (*Figure 2*). As a positive control, we acquired pCF data for 35SGFP, which had a MI = 0.71 ± 0.07 (n = 15). As a negative control we used 3xGFP, which was shown to restrict free GFP movement in roots (*Kim et al., 2005*). To this end, we drove the 3xGFP using a root vasculature promoter (TMO5:3xGFP, *Schlereth et al., 2010*) which had a MI = 0.26 ± 0.05 (n = 19) (*Figure 2—figure supplement 1*).

We then used pCF to measure SHR movement across the vasculature, endodermis, and cortex. We found that SHR moves from the vasculature to the endodermis (MI = 0.58 ± 0.03, n = 20, Wilcoxon with Steel-Dwass, p = 0.6356), consistent with previous experimental data (*Helariutta et al., 2000*; *Nakajima et al., 2001*) (*Figure 2*). However, from the endodermis back to the vasculature, the MI of SHR was significantly lower than the MI of free GFP (MI = 0.36 ± 0.07, n = 20, Wilcoxon with Steel-Dwass, p = 0.0368) and not significantly different from the MI of the 3xGFP (Wilcoxon rank-sum test, p = 0.5974), suggesting that SHR is unable to move back to the vasculature (*Figure 2*). In addition, we found that SHR moves bidirectionally between the endodermis and cortex, as the MI is not significantly different from that of free GFP (MI = 0.61 ± 0.06 from endodermis to cortex, MI = 0.53 ± 0.06 from cortex to endodermis, n = 20, Wilcoxon with Steel-Dwass, p = 0.9524 and p = 0.3909 respectively).

Given that our RICS analysis showed that the presence of SCR reduces SHR mobility in the endodermis, we measured SHR intercellular movement with pCF in the absence of SCR. Accordingly, we performed a pCF analysis using SHR:SHR-GFP in the SCRi line. Line scan measurements were taken

**Table 1.** Recommended imaging conditions for RICS and N&B.

| Method | Pixel size (μm) | Pixel dwell time (μs) | Line scan time (ms) | Number of frames | Image size | Laser intensity | Gain |
|---|---|---|---|---|---|---|---|
| RICS | 0.05 to 0.1 | 12.61 or 25.21 | 7.56 or 15.13 | 50 to 100 | 256x256 | 1.0% to 4.0% | 800 to 1000 |
| N&B | 0.1 | 12.61 or 25.21 | 7.56 or 15.13 | 50 to 100 | 256x256 | 1.0% to 12% | 800 to 1000 |

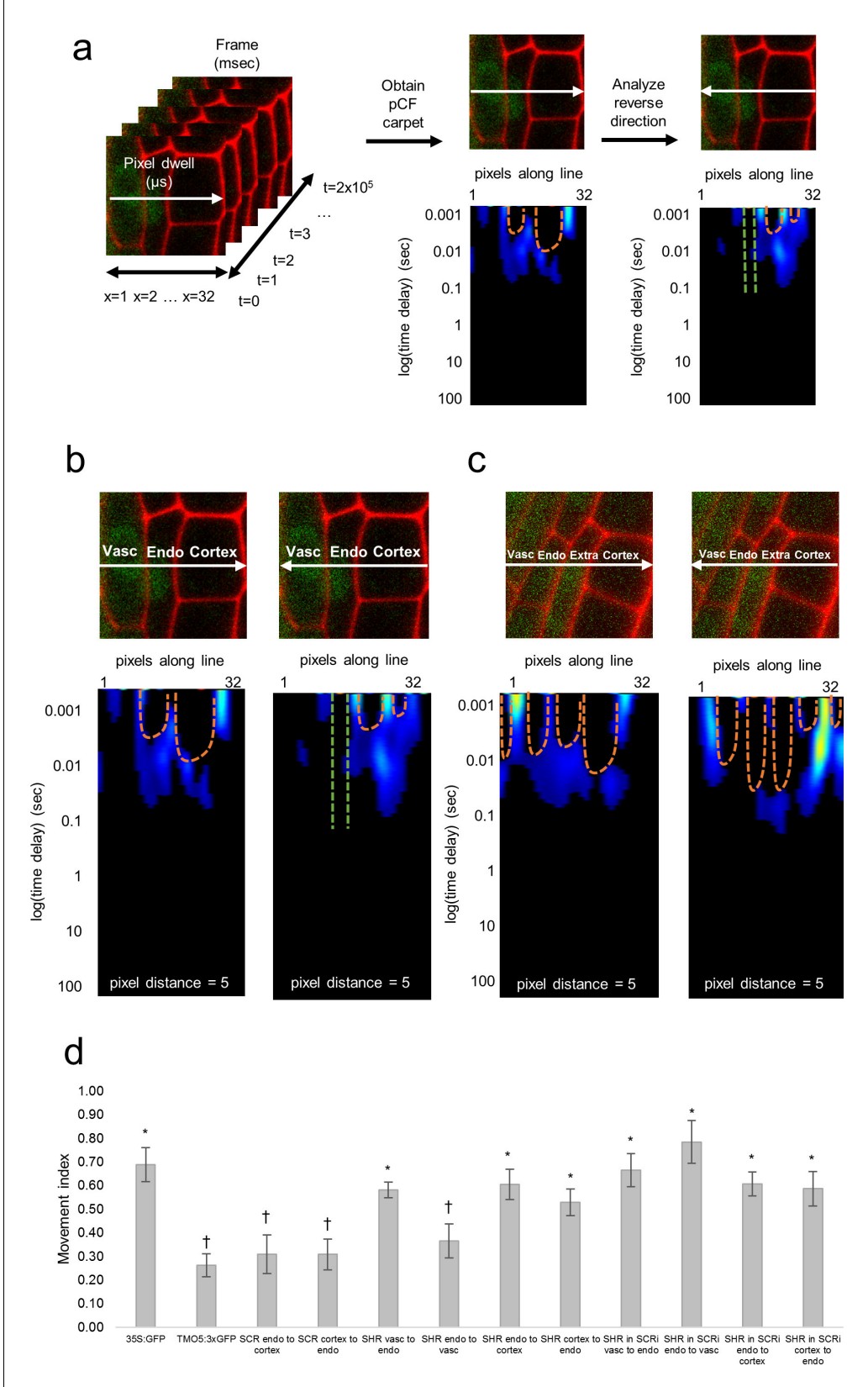

**Figure 2.** Pair correlation function (pCF) analysis showing direction of SHR movement. (a) Schematic of image acquisition and pCF analysis. (Left) Line scans acquired using predetermined imaging conditions (*Table 1*). Carpets of the forward (middle) and reverse (right) pCF analysis. The orange arch

*Figure 2 continued on next page*

*Figure 2 continued*

indicates delayed movement, while the absence of an arch (green lines) indicates no movement. (b) pCF analysis of SHR:SHR-GFP in *shr2*. Cell walls are marked with PI. Lines indicate the laser path going across the vasculature, endodermis, and cortex. pCF carpets for each direction are shown. Orange arches indicate movement. (c) pCF analysis of SHR:SHR-GFP in SCRi. Cell walls are marked with PI. Lines indicate the laser path across the vasculature, endodermis, the extra layer, and the cortex. pCF carpets for each direction are shown. Orange arches indicate movement. (d) Bar graph showing average movement index of 35S:GFP (n = 15), TMO5:3xGFP (n = 19), SCR:SCR-GFP (n = 14), SHR:SHR-GFP in *shr2* (n = 20) between vasculature and endodermis, n = 22 between endodermis and cortex), and SHR:SHR-GFP in SCRi (n = 14 between vasculature and endodermis, n = 17 between endodermis and cortex). Stars denote groups that are different from TMO5:3xGFP, crosses indicate groups that are different from 35S:GFP (Wilcoxon with Steel-Dwass, p<0.05). Error bars are s.e.m. Source data is provided in *Figure 2—source data 1* and *2*.

The following source data and figure supplement are available for figure 2:

**Source data 1.** pCF of 35S:GFP, TMO5:3xGFP, SCR:SCR-GFP, SHR:SHR-GFP in *shr2*, and SHR:SHR-GFP in SCRi lines.
**Source data 2.** Statistical analysis of movement index obtained by pCF.
**Figure supplement 1.** Pair correlation function analysis of 35S:GFP, SCR:SCR-GFP, and TMO5:3xGFP.

across vascular, endodermal, and cortical cells as well as across the extra layer between the endodermis and cortex which is a direct consequence of the RNAi reduction of SCR (*Cui et al., 2007*). We not only observed SHR movement from the vasculature to the endodermis with a MI = 0.66 ± 0.07 (n = 14) but also from the endodermis back to the vasculature with a MI = 0.78 ± 0.09 (n = 14) (*Figure 2*). Since the MI for SHR:SHR-GFP in SCRi in both directions is not significantly different from the MI for 35S:GFP (Wilcoxon rank-sum test, p = 0.9553 for endo to vasc, p = 0.9999 for vasc to endo) it suggests that movement is unrestricted in both directions (*Figure 2*). Similarly, we found that SHR is still able to move bidirectionally between the endodermis and cortex (MI = 0.61 ± 0.05 from endodermis to cortex, MI = 0.59 ± 0.07 from cortex to endodermis, n = 17, Wilcoxon with Steel-Dwass, p = 0.9997 and p = 0.5510 respectively). These results suggest that the presence of SCR prevents SHR movement specifically from the endodermis back to the vasculature while it does not affect the movement between the endodermis and cortex. To determine if SCR can move with SHR from the endodermis to the cortex, we used pCF on the SCR:SCR-GFP. Our results show that SCR does not move between the endodermis and the cortex, as the MI in either direction is significantly lower than that of 35S:GFP (MI = 0.31 ± 0.08 from endodermis to cortex, MI = 0.31 ± 0.07 from cortex to endodermis, n = 14, Wilcoxon with Steel-Dwass, p = 0.0236 and p = 0.0054 respectively). Taken together, our pCF results provide information about the directionality of SHR movement and indicate that SCR restricts SHR movement from the endodermis to the vasculature.

## Protein oligomeric state determined by number and brightness (N&B) analysis

Stoichiometry is an important feature of protein complexes, as some transcription factors must form higher order complexes in order to function (*Nakashima et al., 2012*; *Sornaraj et al., 2016*). We used the Number and Brightness technique (N and B), which relies on the RICS image acquisition, to investigate the oligomeric state of the SHR protein in different root cell types. We used the average fluorescence intensity, and the variance in fluorescence, to determine the brightness of particles and their number in an image (*Digman et al., 2008*; *2009a*) (*Figure 3*, see Materials and methods). In order to use N&B to measure SHR oligomeric state, we first obtained the brightness of the autofluorescence (immobile fraction) and of monomeric GFP protein. We used roots expressing 35SGFP to calculate the S-factor, an imaging parameter that shifts the brightness of the image, such that the immobile fraction has a brightness value of 1 (*Digman et al., 2008*). In addition, we used the 35S:GFP line to measure the brightness of monomeric GFP protein (*Figure 3* and *Table 2*) (see Materials and methods).

After calibration, we used N&B on SHR:SHR-GFP to determine the oligomeric state of SHR in different cell types. Our results indicated that SHR exists primarily as a monomer in the vasculature, given that SHR homodimer formation was only 2.6% ± 0.3% in these cells (*Figure 3*). Notably, SHR monomeric proteins are observed in both the nuclei and in the cytoplasm of vasculature cells. By contrast, in the nuclei of the endodermis, there are significantly more SHR homodimers present

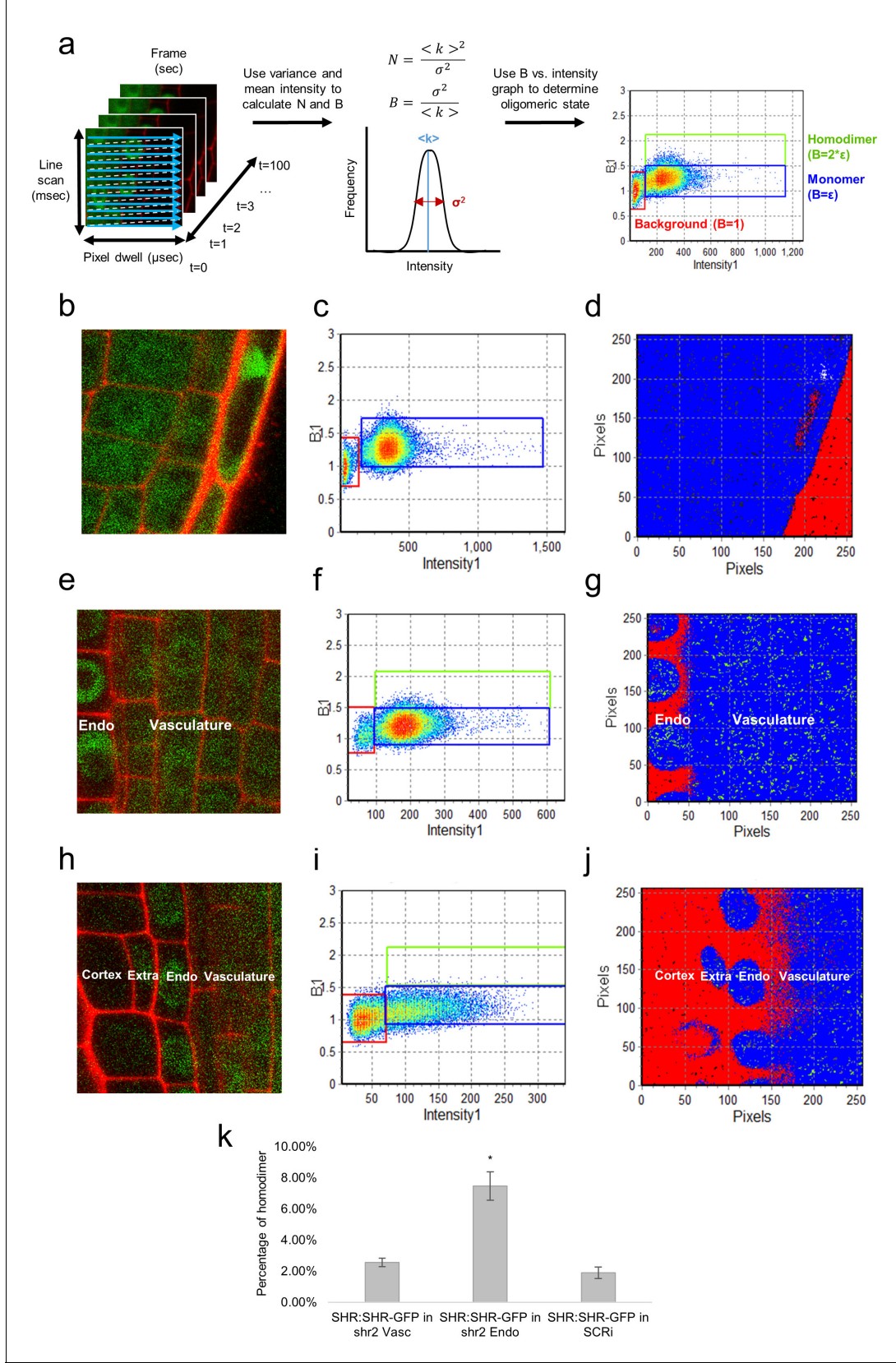

**Figure 3.** N&B analysis of the SHR oligomeric state. (a) Schematic of image acquisition and N&B analysis. (Left) Image acquisition for N&B is the same as for RICS analysis. (Middle) The mean and variance of intensity used to calculate the brightness and number of particles. (Right) The background

*Figure 3 continued on next page*

*Figure 3 continued*

brightness (red) set to 1 by adjusting the S-factor (**Table 2**). The monomer (blue) positioned at the predetermined brightness of monomeric GFP (**Table 2**). Homodimer (green) particles shown to be twice as bright as the monomer. (**b, c, d**) 35S:GFP used to determine the molecular brightness of monomeric GFP (**b, e, h**) Region of interest selected for N&B analysis of 35S:GFP, SHR:SHR-GFP in *shr2*, and SHR:SHR-GFP in SCRi. Cell walls are marked with PI. Note that the extra layer in (**h**) is a result of the SCRi background. (**c, f, i**) Brightness vs intensity for 35S:GFP, SHR:SHR-GFP in *shr2*, and SHR:SHR-GFP in SCRi. The red, blue, green boxes indicate the autofluorescence (B = 1), monomer (B = ε = 0.28 ± 0.01) and homodimer (B = 2*ε), respectively. (**d, g, j**) Color-coding of the brightness for 35S:GFP, SHR:SHR-GFP in *shr2*, and SHR:SHR-GFP in SCRi. Red, blue, and green represent background (autofluorescence), monomer, and homodimer, respectively. (**k**) Bar graph showing average percent of SHR homodimer for SHR:SHR-GFP in vascular cells (n = 40), SHR:SHR-GFP in endodermal cells (n = 19), and SHR:SHR-GFP in SCRi (n = 14). Error bars are s.e.m. Star denotes sample that is significantly different from the other two (Wilcoxon with Steel-Dwass, p<0.05). Source data is provided in *Figure 3—source data 1–3*.

The following source data is available for figure 3:

**Source data 1.** Oligomeric state of SHR:SHR–GFP in *shr2* line obtained using N&B with the Zeiss 780 and Zeiss 710 instruments.
**Source data 2.** Oligomeric state of SHR:SHR–GFP in SCRi line obtained using N&B with the Zeiss 780 instrument.
**Source data 3.** Statistical analysis of the oligomeric state of SHR collected using N&B.

(7.5% ± 0.9% homodimer, Wilcoxon with Steel-Dwass, p<0.0001) (**Figure 3**). Moreover, in the endodermis SHR is present at very low levels outside of the nuclei (**Figure 3**). To further understand the differences in SHR oligomeric state, we determined if the homodimer of SHR present in the endodermis could be affected by the presence of SCR. Therefore, we performed N&B on SHR:SHR-GFP in the SCRi line and observed that SHR is mainly found as a monomer in the endodermis (1.9% ± 0.4% homodimer, Wilcoxon with Steel-Dwass, p = 0.2546), indicating that SCR influences the oligomeric composition of SHR (**Figure 3**). These N&B results provide a quantitative assessment of SHR's oligomeric states and their distribution in the vasculature and endodermis.

## Stoichiometry of the SHR-SCR complex

The N&B analysis revealed that SHR exists both as a monomer and as a homodimer in the endodermis. We next asked if both the monomer and homodimer are able to form a complex with SCR. To test this hypothesis, we performed cross-N&B, which can determine the stoichiometry of the SHR-SCR complex. Cross N&B requires that each protein be tagged with a different fluorophore. The

**Table 2.** N and B parameters for SimFCS software analysis. SEM is given.

| Confocal model and objective | S-factor (green channel) | S-factor (red channel) | Monomer brightness (green channel) (counts/pixel dwell/molecule) | Monomer brightness (red channel) (counts/pixel dwell/molecule) | Cursor size |
|---|---|---|---|---|---|
| LSM 780, 40 x 1.2 NA water | 1.34 ± 0.02 (n = 17) | 1.00 ± 0.01 (n = 24) | 0.28 ± 0.01 (n = 13) | 0.34 ± 0.02 (n = 7) | 42 ± 0.69 (n = 13) |
| LSM 710, 40 x 1.2 NA and 63 x 1.2 NA water | 0.92 ± 0.004 (n = 20) | N/A[*] | 0.24 ± 0.01 (n = 20) | N/A[*] | 50 (n = 20) |

Source data is provided in *Figure 2—source data 1* (Monomer brightness for green channel and cursor size); *Table 2—source data 2* (S-factor, green channel); and Table 2—source data 3 (Monomer brightness and S-factor, red channel)
*Red channel data was not collected on the LSM 710

**Source data 1.** Monomeric brightness of 35S:GFP line obtained using N&B with the Zeiss 780 and Zeiss 710 instruments.
**Source data 2.** S-factor of the 35S:GFP background line obtained using N&B with the Zeiss 780 and Zeiss 710 instruments.
**Source data 3.** S-factor of the UBQ10:mCherry background line and monomeric brightness of UBQ10:mCherry line obtained using N&B with the Zeiss 780 instrument.

analysis then determines which proteins are in a complex by calculating the cross-correlation between the two channels at each pixel (*Figure 4*, see Materials and methods). Accordingly, we generated a transgenic line containing both SHR and SCR tagged with different fluorophores (SHR:SHR-GFP & SCR:SCR-mCherry) (*Figure 4—figure supplement 1*). We reasoned that SCR may also exist in higher oligomeric states, which would increase the possible binding ratio of the SHR-SCR complex; therefore we first used N&B on SCR:SCR-mCherry to determine the oligomeric state of SCR. Initially, we determined the S-factor and brightness of monomeric mCherry protein in the root (UBQ10:mCherry) as we did for the monomeric 35SGFP protein (*Figure 4—figure supplement 2* and *Table 2*). In the SCR:SCR-mCherry line, we detected mostly monomers with 4.7% ± 0.5% of homodimers (*Figure 4*). We tested the oligomeric state of SCR using a SCR:SCR-GFP fusion protein (*Figure 4—figure supplement 2*), as it was shown that the type of fluorescent tag can change the behavior of a protein and its aggregation (*Brown et al., 2008*). We found that the SCR:SCR-GFP line had 5.4% ± 0.7% of homodimers, which is similar to the SCR:SCR-mCherry line (*Figure 4—source data 1*). Therefore, when performing the cross-N&B analysis, we considered the possibility that the homodimer of SCR could be part of the SHR-SCR complex.

After analyzing the SHR:SHR-GFP and SCR:SCR-mCherry separately, we performed cross-N&B on the SHR:SHR-GFP/SCR:SCR-mCherry line to determine the stoichiometry of the complex. The cross-N&B analysis returns a stoichiometry diagram that represents the proportion of different complexes (*Figure 4*). We found that both the monomer and the homodimer of SHR bind the monomer of SCR, suggesting that SHR and SCR bind with a 1:1 and 2:1 stoichiometry (*Figure 4*). We were not able to detect any complexes that contain the homodimer of SCR. Additionally, we determined that 84.8% ± 1.6% of the SHR-SCR complexes have 1:1 stoichiometry while 15.2% ± 1.6% have 2:1 stoichiometry (*Figure 4*). Our cross N&B results reveal that both the monomer and homodimer of SHR are able to bind the monomer of SCR, while the homodimer of SCR does not seem to be part of this complex.

## A mathematical model of SHR and SCR dynamics

SHR and SCR dynamics have been previously modeled in the endodermis, but this model did not take into account either the intercellular movement of SHR or the stoichiometry of the SHR-SCR complex (*Cruz-Ramırez et al., 2012*). After experimentally determining the rate of SHR movement, SHR oligomeric state, and the binding ratio of the SHR-SCR complex, we sought to incorporate this information into a mathematical model of SHR and SCR, with the goal of determining how protein movement and stoichiometry affect SHR and SCR dynamics. Therefore, we constructed a compartmental model that only measures SHR, SCR, and the SHR-SCR complex in both the vasculature and the endodermis (see Materials and methods).

After constructing the model, we performed a sensitivity analysis to determine the most influential parameters. We chose to use Sobol indices to measure how sensitive the model is to each parameter (*Sobol, 2001*; see Materials and methods). Briefly, the total Sobol effects index measures how much the model outcome varies as the parameters are changed. If small changes in the parameter values cause large changes in the model outcome, then that parameter is more influential. Using this measure, we found that the rate of movement of SHR from the vasculature to the endodermis ($a_1$) and from the endodermis to the vasculature ($a_2$) are both highly influential parameters (*Figure 5—figure supplement 1*). Thus, this suggests that SHR movement is a key component of the model that greatly influences the dynamics of SHR in the endodermis.

Next, we sought to use the model, in conjunction with our experimentally determined diffusion coefficients, to (i) simulate SHR and SCR dynamics in the endodermis, and (ii) estimate values for the other parameters. Most of the parameter values were chosen based on the previous mathematical model (*Cruz-Ramírez et al., 2012*). However, since this model did not account for different oligomeric states of SHR we estimated some of the parameters using the N&B data (*Supplementary file 3*; see Materials and methods). Our mathematical model predicts that SHR reaches a steady state in the vasculature and endodermis in a matter of minutes. The levels of SCR increase greatly in the first 3 hr, which is supported by data that show that SCR expression in a SHR inducible system is significant after 3 hr (*Sozzani et al, 2010*). While the 1:1 SHR-SCR complex increases greatly in the first 3 hr, the 2:1 complex does not form until after 9 hr. This is because the SHR homodimer does not form until about 9 hr into the simulation (*Figure 5*). We reasoned that this is a plausible scenario because SCR should exceed 60% of steady-state levels to trigger homodimer formation (see Materials and methods).

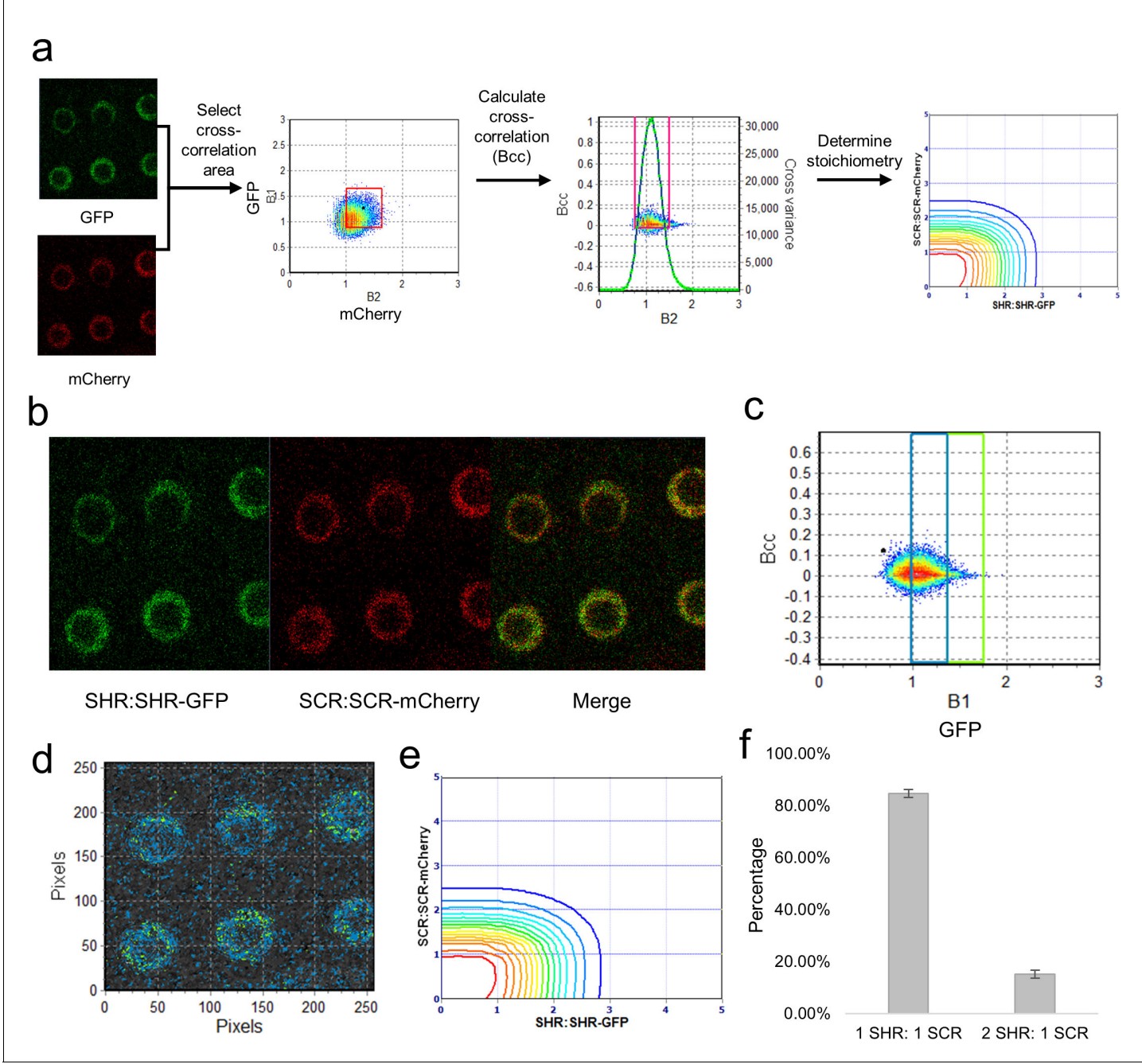

**Figure 4.** Cross-N&B analysis of a SHR/SCR double-tagged line. (**a**) Schematic of cross N&B analysis. (Left) A double-tagged line used for imaging. The B1 (GFP brightness) vs B2 (mCherry brightness) graph is used to select the region for cross-correlation. (Middle) The brightness cross-correlation (Bcc) used to determine GFP pixels that cross-correlate with mCherry pixels. (Right) Stoichiometry plot that displays the protein complexes detected in the image. (**b**) Expression of SHR:SHR-GFP/SCR:SCR-mCherry marker line in root endodermis. (**c**) Bcc vs B1 graph for SHR. The blue and green boxes represent the SHR monomer and homodimer, respectively, that form a complex with SCR. (**d**) Color-coding of the cross brightness of the SHR:SHR-GFP/SCR:SCR-mCherry line. Blue represents SHR monomer binding SCR monomer, while green represents SHR homodimer binding SCR monomer. (**e**) Stoichiometry histogram from cross N&B analysis. The orange line at (1,1) represents a high proportion of monomeric SHR bound to monomeric SCR (84.77% ± 1.58%), while the green line at (2,1) represents a lower proportion of homodimeric SHR bound to monomeric SCR (15.23% ± 1.58%). (**f**) Bar graph showing average percentages of the 1:1 and 2:1 SHR-SCR complex (n = 17). Error bars are s.e.m. Source data is provided in *Figure 4—source data 1* and *2*.

The following source data and figure supplements are available for figure 4:

**Source data 1.** Oligomeric state of SCR:SCR-GFP and SCR:SCR-mCherry lines obtained using N&B with the Zeiss 780 instrument.

*Figure 4 continued on next page*

*Figure 4 continued*

**Source data 2.** Stoichiometry of the SHR:SHR-GFP/SCR:SCR-mCherry complex obtained using cross N&B with the Zeiss 780 instrument.

**Figure supplement 1.** Longitudinal confocal root sections of SHR:SHR-GFP/SCR:SCR-mCherry line.

**Figure supplement 2.** N&B analysis of UBQ10 and SCR oligomeric state.

Finally, the entire system reaches a steady-state between 18 and 24 hr (*Figure 5*). This suggests that cell division occurs once SCR and the SHR-SCR complexes reach their steady state values.

In addition to simulating what happens in wild type, we also wanted to observe how decreasing SCR affects the model dynamics and how it reflects our pCF and N&B data. Accordingly, in the SCRi model, SHR movement is bidirectional and no SHR homodimer forms (see Materials and methods). In this model, we assume that SCR levels are maintained at below 60% of wild type levels as in agreement with previous experimental data (*Cui et al, 2007*). At these lower levels of SCR, we observe that the steady-state value of the 1:1 SHR-SCR complex is reduced by half (*Figure 5*). In addition, we observe that SHR homodimers do not form as seen in the N&B analysis (*Figure 3*). Since SHR homodimers do not form, the 2:1 SHR-SCR complex does not exist (*Figure 5*). These results suggest that the extra layer in the SCRi line could form due to a reduction in the levels of the SHR-SCR complex. However, there are still other unknown factors that could contribute to the formation of the extra layer.

## Discussion

The methodologies we describe are rapid, direct, and convenient approaches for characterizing the quantitative and qualitative behavior of proteins in vivo. Combining RICS and pCF provided detailed understanding of SHR movement both within and between cells of the *Arabidopsis* root. Trafficking along the radial axis is normally unidirectional - SHR moves from the vasculature to the endodermis, and not in the opposite direction (*Figure 2*). In the absence of SCR, SHR trafficking is bidirectional. In agreement with the pCF results, our RICS data showed that the DC of SHR in the endodermis increases significantly in the absence of SCR as compared to wild type, indicating that removal of SCR affects the diffusive behavior of SHR. Taken together, these results provide further evidence that SCR, which is present only in the endodermis, spatially restricts SHR movement. The mechanisms underlying this restricted movement remain unclear, and future experiments should focus on uncovering other factors, in addition to SCR, that can bind SHR (*Long et al., 2015*; *Moreno-Risueno et al., 2015*). In this study, combining RICS and pCF provided quantitative information about the speed and direction of protein movement.

Using RICS and pCF, which allow both intra- and inter-cellular analysis of protein movement, we were able to determine the in vivo flow of molecules in a multicellular organism. The average value of the diffusion coefficient obtained with RICS could be obtained with other techniques such as single-point FCS or Fluorescence Recovery After Photobleaching (FRAP), but these techniques have limitations, such as returning only temporal information (*Miyawaki, 2011*). On the other hand, both RICS and pCF are applied to frames or lines, respectively, which increase the spatial resolution and the statistical power of the analysis by simultaneously measuring the movement of many individual molecules in time and space. Moreover, RICS and pCF introduce a spatial component that can determine whether slow movement is due to a binding interaction or slow diffusion, which FRAP and single-point FCS cannot do. Thus, these scanning FCS methods are robust against cell movement and other artifacts that could bias measurements taken with other single-point methods.

The DC measurements of SHR in both the *shr2* complemented line and SCRi are significantly lower than those of freely-moving 35S:GFP (Wilcoxon rank-sum test, p<0.0001) (*Figure 1*). The slower movement of SHR could be due to its size alone. However, since the DC of SHR is three-fold smaller than the DC of 35S:GFP, and the DC is inversely proportional to the cube root of the molecular weight of the protein (*Young et al., 1980*), the change in DC is not attributable to the molecular weight of SHR (~60 kDa). If it were, then the molecular weight of SHR would have to be 27-fold

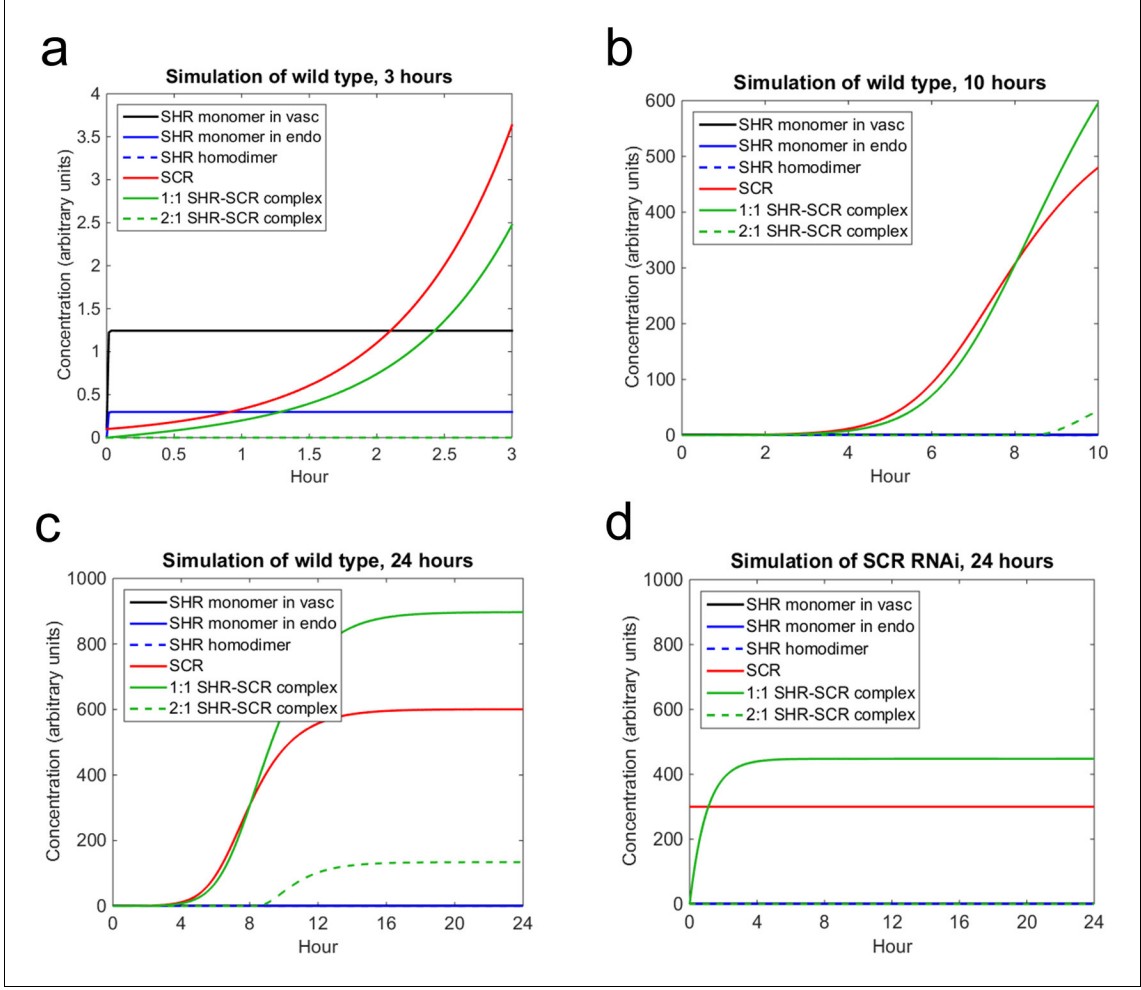

**Figure 5.** Mathematical model simulations of SHR and SCR illustrate how reduction of SCR affects the formation of SHR homodimer and SHR-SCR complex. (a, b, c) Model simulations of wild type showing how (a) SCR and the 1:1 SHR-SCR complex greatly increase in the first 3 hr, (b) SHR homodimer and the 2:1 SHR-SCR complex do not form until around 9 hr, (c) the entire system reaches a steady state between 18–24 hr. (d) Model simulations of SCR RNAi showing a reduction in SHR homodimer, SCR, 1:1 SHR-SCR complex, and 2:1 SHR-SCR complex levels after 24 hr. The model outcomes show SHR in the vasculature (black), SHR monomer in the endodermis (solid blue), SHR homodimer (dashed blue), SCR (red), 1:1 SHR-SCR complex (solid green), and 2:1 SHR-SCR complex (dashed green). Parameter values and initial conditions are given in *Supplementary file 3*. Source data is provided in *Figure 5—source data 1* and *2*.

The following source data and figure supplements are available for figure 5:

**Source data 1.** Sobol total effects indices computed for SHR-SCR mathematical model.

**Source data 2.** Area measurements of vascular and endodermal cells.

**Figure supplement 1.** Sensitivity analysis of mathematical model of SHR and SCR.

**Figure supplement 2.** Functional form of $k_2$ parameter in mathematical model.

higher than that of GFP, or approximately 729 kDa (*Prendergast and Manni, 1978*). Further, the decrease in the DC of SHR relative to 35S:GFP, in addition to the pCF analysis which shows unidirectional movement, suggests an active regulation of SHR movement, which is in agreement with previous data (*Gallagher et al., 2004*; *Sena et al., 2004*; *Gallagher and Benfey, 2009*). Future work could expand our understanding of SHR movement by examining it in callose synthase gain-of-function mutants, which block transport via the plasmodesmata (*Vaten et al., 2011*), or in Shortroot

interacting embryonic lethal (SIEL) mutant alleles, which have reduced SHR movement (*Koizumi et al, 2011*).

The N&B results provided a quantitative assessment of SHR's oligomeric states and their distribution across cell types. SHR exists in the vasculature primarily as a monomer and in the endodermis as a monomer and a homodimer (*Figure 3*). The presence of the SHR homodimer in the nucleus and loss of the homodimer in the SCRi line indicate that SCR is required to maintain SHR in its higher oligomeric forms. Taken together, these results provide insights into the molecular mechanisms by which SCR regulates SHR movement. They also raise new questions, such as how SCR maintains SHR in multimeric forms, and how formation of higher oligomeric complexes helps restrict SHR movement.

By using our experimentally determined parameters in a mathematical model, we were able to determine that SHR reaches a steady state in a matter of minutes, while SCR and the SHR-SCR complex stabilize within 24 hr. This suggests that future experiments that aim to understand the details of the SHR-SCR GRN should focus on a time scale of hours to measure its dynamics. In addition, the sensitivity analysis revealed that diffusion is one of the most important parameters in our model, motivating the need to experimentally measure the diffusion coefficient. In the simulation of SCRi, our model showed that the 2:1 complex does not form and the 1:1 complex is reduced to half of normal levels. This suggests that the mutant layer in the SCRi line is likely due to a reduction in the SHR-SCR complex. These results only scratch the surface of what is likely to be a complex network regulating the spatial localization of SHR as a mobile transcription factor and developmental regulator. Furthermore, they highlight the importance of physical interactions between transcription factors as a regulatory component of transcription factor intercellular trafficking networks. The three scanning FCS methodologies utilized here (pCF, RICS, and N&B) gave consistent results providing strong evidence for their reliability. Our data suggest that the application of in vivo molecule tracking techniques is virtually limitless, which opens exciting new opportunities in all fields of biology.

## Materials and methods

### Plant material and growth conditions

Prior to plating, *Arabidopsis* seeds were dry sterilized using 50% bleach and 1.5 ml of HCl for at least 1 hr, imbibed with 500–700 µL of sterile water, and vernalized for 2 days at 4°C in complete dark. After vernalization, seeds were plated on 1x MS (Murashige and Skooge) media supplemented with 1% sucrose and grown vertically at 22°C in long day conditions (16 hr light/8 hr dark). Seedlings were 5 days old when imaged. The 35SGFP, SHR:SHR-GFP in *shr2*, SCR:SCR-GFP in *scr4*, SHR:SHR-GFP in SCRi, and TMO5:3xGFP lines are described in (*Ruiz et al., 1998*; *Nakajima et al., 2001*; *Cui et al., 2007*; *Sabatini et al., 2003*; *Schlereth et al, 2010*). For the SCR:SCR-mCherry construct, the coding sequence (CDS) of the *mCherry* fluorescent protein (*Goedhart et al., 2007*) was amplified using primers with flanking *att*B sites: mCherry-R2R3 F: 5'-GGGGACAGCTTTCTTGTACAAAG TGGCTATGGTGAGCAAGGGCGAGGAG-3' and mCherry -R2R3 R: 5'-GGGGACAACTTTGTATAA TAAAGTTGCTTACTCACTTGTACAGCTCGTCCATGCC-3 and recombined into pGEMTeasyR2R3 vector by Gateway BP reaction. *SCR* coding sequence in pDONR221-derived entry clones was previously described (*Welch et al., 2007*). The root expression vector was created using endogenous SCR promoter in pH7m34GW binary vectors by multiple Gateway LR reactions as described (*Long et al., 2015*).

### Statistical analysis

We performed all statistical analyses using the Wilcoxon rank-sum test with Steel-Dwass for multiple comparisons at significance level $\alpha = 0.05$. We chose the Wilcoxon test as not all of our data are normally distributed (Shapiro-Wilk Goodness of Fit test, $p < 0.0001$, *Supplementary file 1*). In addition, the Wilcoxon test adjusts for our sample sizes, which were unequal between groups. The Steel-Dwass test is used after the Wilcoxon test to perform pairwise comparisons. All statistical analyses were performed using JMP software (http://www.jmp.com).

### Point spread function (PSF) measurement

The Point Spread Function (PSF) measures the radius of the laser beam and is experimentally measured in order to perform Raster Image Correlation Spectroscopy (RICS) analysis (*Rigler et al.,*

*1993*). The PSF was measured for each objective and each confocal microscope used for image acquisition. To calibrate the PSF for our objective lens, we performed RICS on free EGFP in an aqueous solution. While we determined the PSF using free EGFP in solution, it is possible that the PSF varies depending on the cell or tissue we are imaging. However, the exact value of the PSF is not as important for these scanning FCS techniques as it is for single-point FCS since the scanning techniques measure the time a molecule takes to go from one pixel to the next (*Digman et al., 2005b*; *Petrasek and Schwille, 2008*). Thus we made the assumption that the PSF determined using the EGFP solution is an accurate estimation of the true PSF. Accordingly, raster images of a solution of 0.6 µM EGFP were acquired using commercial CLSMs, including the Zeiss 780 and 710 (Zeiss Inc, Germany). We determined the PSF by fitting the autocorrelation function (ACF) to the intensity fluctuations of free EGFP in aqueous solution, obtained from the raster images, while fixing the known EGFP diffusion coefficient of 78 µm$^2$/s (*Chen et al., 2002*). The ACF then returned the experimental PSF beam waist (*Supplementary file 2*). We found that the PSF measurement was different for different objectives (*Supplementary file 2*).

## Raster image correlation spectroscopy (RICS)

RICS is a technique that has been developed to determine the rate of movement (either passive diffusion or regulated movement) of fluorescently-labeled particles in a small volume (i.e femtoliter volume of the PSF). In particular, RICS can be used to determine the rate of movement of a GFP-labeled protein, such as SHR:SHR-GFP. During a RICS measurement, the laser performs a raster scan across a selected region of interest. The raster scan involves scanning from left to right across a set number of pixels and then retracing, without backtracking across the already imaged pixels, to perform the next line below. The scanning is repeated for multiple consecutive lines until a 256x256 pixel frame is created, and then starts over again to obtain between 50 to 100 time points. Because each pixel is collected at a different time, and we know where each pixel is located, there is spatial and temporal information included in each individual image (*Figure 1*). This spatio-temporal information can be used to extract the diffusion coefficient of a population of molecules using the RICS algorithm, which has two steps: (1) background subtraction and (2) image correlation. The background subtraction removes stationary and slow-moving objects so that the image autocorrelation only detects the dynamics of diffusing species (*Digman et al., 2005a*) while the image correlation compares each pixel to its adjacent pixel in both the x and y direction.

For the RICS analysis the pixel size, pixel dwell time, line scan time, number of frames, and image size were set prior to acquisition. The parameters were set so that each pixel is sampled at a slightly faster rate than the particles move in the solution (*Digman et al., 2005a*). The laser intensity was set so that the signal to noise ratio is high, but within a range that does not cause photobleaching (*Table 2*). Since all of these parameters affect the ACF fit and, therefore, the estimation of the diffusion coefficient, we determined the optimal parameters for the *A. thaliana* root system by testing different values while imaging the 35S:GFP genotype. We chose parameter values that resulted in an ACF with a good fit to the data and low residuals (*Table 1*). We found that these parameter values remained the same between different confocal microscopes. Moreover, since sample movement can create artifacts in the ACF, which results in an erroneous diffusion coefficient (*Digman et al., 2005a*), no data were analyzed that had significant sample movement (i.e. sample movement that shifts the region of interest outside of the imaging frame).

The RICS-ACF is decomposed into two correlation functions that depend on $\xi$ (the spatial lag in x) and $\psi$ (the spatial lag in y). The first correlation function, $S(\xi,\psi)$, calculates the spatio-temporal correlation due to the scanning of the microscope. The second correlation function, $G(\xi,\psi)$, calculates the spatio-temporal correlation due to particles diffusing in the medium. The ACF, $G_S(\xi,\psi)$, takes both of these correlations into account by multiplying them: $G_S(\xi,\psi)= S(\xi,\psi)* G(\xi,\psi)$. The functions are constructed assuming that the distribution of fluorescence intensities follows a 3D Gaussian distribution. The decomposition of the ACF into two parts allows RICS to distinguish random, Brownian motion from diffusing particles in the medium (*Digman et al., 2005a*).

It is possible for artifacts to occur in the ACF due to slow, mobile structures within the volume. To eliminate these artifacts from our analysis, a moving average with a 10 frame time window was applied to the time series images to subtract the immobile fraction (*Digman et al., 2005a*). The first and last 5 frames of the time series were ignored since in this time window there is not enough information to calculate the moving average. For the remainder of the frames, the average intensity of

the 5 preceding and 5 following frames was subtracted. For example, the average of frames 10–20 was subtracted from frame 15.

The ACF was calculated using the PSF of the beam (*Supplementary file 2*) and the other parameters that were set before imaging (*Table 1*). The only unknown parameters in the ACF function were DC, the diffusion coefficient of the particles, and G(0), which is inversely proportional to the number of molecules present. The diffusion coefficient returned by the software was the value that best fits the data (see SimFCS Software Analysis).

## Pair correlation function

Pair correlation function (pCF) is a technique that allows us to measure the movement of a protein along a line. To do this, we scanned a 32 pixel line through a region of interest in the root. The region of interest was chosen such that 2–3 cells are contained within the frame. To visualize cell walls, we used propidium iodide (PI), which causes the walls to fluoresce red when excited with the 488 nm emission line of an argon laser. In our images, the line scans across the middle of the frame, and the sample was positioned such that the line does not scan directly over a cell wall. Additionally, the line was placed towards the middle of the cell so that movement in both the nucleus and cytoplasm can be measured. The imaging area was cropped such that the edges of the line overlap exactly with the outer cell walls. Once the imaging area was cropped, a reference image was taken to check for image movement. The selected 32 pixel line was then rapidly scanned $2 \times 10^5$ times at a pixel dwell time of 12.61 µs. Note that the pixel size was not set as a constant, but rather changed for each image depending on the position of the line scan and the size of the cells analyzed. As soon as the line scan finished, another reference image was taken to check if the root had moved during imaging. If the reference images suggested that the root had moved during imaging, then that line scan was not used for analysis.

The pair correlation function (pCF) for two points at a distance $\delta r$ as a function of the delay time $\tau$ is calculated using *Equation (1)*:

$$G(\tau, \delta r) = \frac{\langle F(t,0).F(t+\tau,\delta r) \rangle}{\langle F(t,0) \rangle \langle F(t,\delta r) \rangle} - 1,$$

(1)

where F(t,0) is the fluctuation in fluorescence intensity at pixel 0 and F(t+τ,δr) is the fluctuation in fluorescence intensity at some other pixel position (δr) at different time delays (τ) (*Hinde et al., 2010*; *Hinde et al., 2011*; *Digman and Gratton, 2009b*). The result of the pCF analysis is a carpet, or heatmap, that displays the correlation in fluorescence over time (y-axis) and space (x-axis). Molecules that move across a barrier display a characteristic 'arch' pattern in the pCF carpet output, whereas molecules that do not move across the barrier do not (*Hinde et al., 2010*) (*Figure 2*). Since the delay time recovered by the pair correlation function analysis is variable, we performed a binary analysis on the pCF carpets to look for movement (presence of an arch pattern) or no movement (no arch) (see SimFCS Software Analysis).

## Number and brightness (N&B)

N&B is used to determine the number (N) and brightness (B) of particles in a volume, which allows us to determine the amount of aggregation of particles. This is useful in determining the oligomeric state of GFP- and mCherry-labeled proteins such as SHR:SHR-GFP, SCR:SCR-GFP, and SCR:SCR-mCherry. A time course of raster-scanned images was obtained using the confocal microscope (see RICS). Certain imaging parameters had to be determined so that the pixels were not under or over sampled (see RICS). In addition, lines expressing monomeric forms of GFP (35S:GFP) and mCherry (UBQ10:mCherry) were used to set the background fluorescence and to measure monomer brightness (see SimFCS Software Analysis).

We obtained both the mean and the variance of the intensity distribution at each pixel in order to determine the number (N) and brightness (B) of the particles. The mean, $<k>$, and the variance, $\sigma^2$, of the intensity distribution are given by *Equations 2 and 3*:

$$<k> = \frac{\sum_{i=1}^{K} k_i}{K}$$

(2)

$$\sigma^2 = \frac{\sum_{i=1}^{K}(k_i - <k>)^2}{K} \tag{3}$$

where $K$ is the number of time points and $k_i$ is the fluorescence counts for time point $i$. The number and brightness of the particles can be determined from the mean and variance of the intensity distribution alone due to the assumption that the occupation of particles follows a Poisson distribution (*Digman et al., 2008*). Using moment analysis (*Qian and Elson, 2000*), the apparent number ($N$) and apparent brightness ($B$) of the particles are defined in *Equations 4 and 5*.

$$N = \frac{<k>^2}{\sigma^2} \tag{4}$$

$$B = \frac{\sigma^2}{<k>} \tag{5}$$

Note that if the average intensity is fixed and the variance increases, $B$ increases but $N$ decreases. This is because fewer, larger particles cause greater intensity fluctuations as the laser scans than many small particles. The true number of particles, $n$, and the true brightness, $\varepsilon$, can be calculated from $N$ and $B$ respectively. However, the apparent brightness $B$ is used for the software analysis (see SimFCS Software Analysis).

## Cross number and brightness (Cross N&B)

Cross N&B follows the same theory as N&B but involves two particles that are marked with different fluorescent proteins (*Digman et al., 2009b*). Thus, Cross N&B is used to determine the binding ratio of a protein-protein interaction. We specifically use Cross N&B to look at the stoichiometry of the SHR-SCR complex using the SHR:SHR-GFP/SCR:SCR-mCherry line. The apparent number of particles, $N$, and the apparent brightness, $B$, were calculated for the green and the red channels separately (see N&B). The cross-variance, $\sigma_{cc}^2$, is defined in *Equation 6*:

$$\sigma_{cc}^2 = \frac{\sum_{i=1}^{K}(G_i - <G>)(R_i - <R>)}{K} \tag{6}$$

where $G_i$ and $R_i$ are the pixel intensities in the green and red channels, respectively, at time $i$, and $<G>$ and $<R>$ are the mean intensities of the green and red channels. When the cross-variance is zero, the fluctuations in the two channels are independent. When the cross-variance is positive or negative, the two channels are correlated or anti-correlated, respectively. The cross-brightness, $B_{CC}$, and the cross-number, $N_{CC}$, are defined in *Equations 7 and 8*.

$$B_{cc} = \frac{\sigma_{cc}^2}{\sqrt{<G><R>}} \tag{7}$$

$$N_{cc} = \frac{<G><R>}{\sigma_{cc}^2} \tag{8}$$

To determine the stoichiometry of a protein-protein complex, the brightness of each protein was compared to the cross-brightness at each pixel (*Digman et al., 2009b*). A large, positive cross-brightness indicates that the two proteins bind at that pixel in the image. The brightness of each protein in the complex determines the stoichiometry (see SimFCS Software Analysis).

## SimFCS software analysis

The SimFCS Software (*Digman et al., 2005a*), developed at the Laboratory for Fluorescence Dynamics (www.lfd.uci.edu), is used to perform RICS, pCF, N&B, and Cross N&B analysis on raster or line scans obtained using a confocal microscope. For the RICS analysis, the software can reduce the region of interest (ROI) from 256x256 pixels to 128 × 128 pixels in order to obtain a more enriched cell population. For example, we used this feature to obtain QC-enriched populations (*Figure 1*). After selecting the ROI to use, the software uses the moving average (see RICS) to eliminate any artifacts from immobile fractions. Then, the software fits the RICS-ACF using the imaging parameters

provided (*Table 1*) and returns the diffusion coefficient of the protein. The diffusion coefficient returned results in the ACF curve that best fits the data (*Figure 1*). Goodness of fit was determined by comparing the residuals to the amplitude of the ACF. We only kept images where the maximum value of the ACF was three fold larger than the greatest residual in order for the RICS analysis to be reliable (*Figure 1*). Images that had residuals larger than this threshold generally had low laser intensity or sample movement were not used for analysis.

For pCF, the line scan file was loaded as a 32 pixel by $2 \times 10^5$ pixel image where the *x* axis represents position along the line and the *y* axis represents time. When there was photobleaching in the sample, we eliminated some of the frames acquired at later time points. A period average of 800 frames was then applied so that trends in the fluorescence carpet are easier to see. The carpet is displayed as a gradient, with red corresponding to high correlation and blue corresponding to low correlation. Next, the autocorrelation of each of the 32 pixels in the line scan was calculated using a moving average of 200 frames. The resulting image was a 32 column image where each of the columns represented the autocorrelation value of that pixel. At this step, the 32 columns were aligned with the reference image to determine the pixel location of the cell wall. The location of the cell wall was then used as the column distance ($\delta r$) in order to calculate the pCF. The software calculated the pCF by correlating pixels that are $\delta r$ apart, moving from left to right. The pCF in the opposite direction was then calculated by correlating pixels from right to left, instead of left to right. To account for the fact that the cell wall is not straight, the pCF was calculated using pixel distances of 5, 7, and 9. Given that there is heterogeneity in the cell size and cell wall orientation in the root, we would not capture differences by using only one pixel distance. Finally, the color scale of the pCF was adjusted such that high correlation is represented as red, low correlation is represented as blue, and no correlation is black.

N&B analysis was applied to the same data set as RICS. First, we calibrated the software using a monomeric form of the fluorescent proteins, namely, 35S:GFP and UBQ10:mCherry. Images were taken of *A. thaliana* roots expressing 35S:GFP and UBQ10:mCherry that contained and did not contain the background (background refers to a region of the image that does not contain the root) using the same experimental settings as the RICS analysis (*Figure 3*). RICS analysis was run on the images to ensure a good ACF fit. Then, N&B analysis was first run on the background images. The software plots the brightness versus the intensity of each pixel. Since the intensity is not used in the N&B calculation, the exact values of the intensity for each fluorophore do not matter. Background images have two distinct populations representing the monomer and the background. The background brightness was standardized by setting the S-factor parameter such that the background population is centered at *B = 1* (*Figure 3*). This ensured that the brightness of the monomer was calibrated for the detector output. Since background brightness can vary between microscopes and laser or detector settings, the S-factor was calculated for each microscope and experimental set-up (*Table 2*).

Once the S-factor was set, N&B analysis was run on the 35S:GFP and UBQ10:mCherry images with no background. All particles with *B > 1* were bounded by a rectangle, or cursor, on the brightness graph, and the position and size of the cursor on the *B* axis were recorded (*Table 2*, *Figure 3*). The size of the cursor measures the distribution of the monomer in the image. The position on the *B* axis represents the brightness of the GFP or mCherry monomer. The x axis represents the intensity and is not used in the N&B analysis: thus, the intensity axis can vary between images. Since these parameters can vary by microscope, the cursor size and monomer brightness were calculated for each microscope (*Table 2*).

The quality of acquired data for N&B analysis of GFP- and mCherry-labeled proteins was first determined by RICS analysis of the acquired images to ensure a good ACF fit. The S-factor was set to the value determined by the background image analysis (*Table 2*). A cursor size was selected that took into account the entire distribution of monomer detected in the brightness histogram. Another cursor of the same size was then positioned at the brightness value that corresponds to twice the brightness of the monomer, as any pixels inside the higher rectangle represent a homodimer of the protein (*Figure 3*). The percentage of monomer and homodimer were then calculated by dividing the number of pixels inside the monomer, or homodimer, cursor by the total number of fluorescent pixels (monomer plus homodimer).

For Cross N&B using images that contain both GFP- and mCherry-labeled protein, the B1-B2 channel was used to determine the pixels in the green channel that were positively correlated with the red channel. The cursor was positioned in the area of the B1-B2 channel that corresponded to where the GFP and mCherry monomers are located. The cursor was then expanded to include any

higher oligomeric states that are present, and that area was set as the correlation area (*Figure 4*). Once the correlation area was set, the Brightness cross correlation (Bcc) channel was used to determine the brightness of the green and red channels at each of the correlated pixels (*Digman et al, 2009b*). The software highlighted the mCherry pixels that correlated with the GFP pixels using a green curve. The percentage of each complex stoichiometry (1:1, 2:1, etc) was then calculated by overlaying cursors on GFP monomer, mCherry monomer, GFP homodimer, and mCherry homodimer. As in N&B analysis, the percentage was computed by dividing the pixels in the 1:1 complex by the total number of fluorescent pixels (1:1 plus 2:1). Finally, the software returned a stoichiometry plot that displayed the most likely stoichiometry of the protein-protein complex (*Figure 4*).

## Mathematical model formulation

We constructed a mathematical model that incorporated our experimentally determined parameters and simulated SHR and SCR dynamics in the root. Our model assumed that transcription and translation happens quickly. Because of this, we modeled transcription and protein movement terms in the same equation. Additionally, we assumed that all proteins have linear degradation terms.

First, we developed a model reflecting wild type conditions (Model 1). We modeled six different variables: SHR in the vasculature ($S_v$), SHR monomer in the endodermis ($S_e$), SHR homodimer in the endodermis ($S_{2e}$), SCR in the endodermis ($C$), 1:1 SHR-SCR complex in the endodermis ($SC$), and 2:1 SHR-SCR complex in the endodermis ($S_2C$). The model consisted of six ordinary differential equations (ODEs) that measure how each of the variables changes over time.

We assumed that SHR is constantly produced at rate $k_1$ as there is no information on upstream regulators of SHR. Since our pCF analysis showed that SHR only moves from the vasculature to the endodermis, possibly through an active transport mechanism (*Gallagher et al., 2004*; *Sena et al., 2004*; *Gallagher and Benfey, 2009*), we modeled the movement of SHR using an active transport term, where $a_1$ is the active transport rate. We defined $a_1$ as the experimentally determined diffusion coefficient ($D_1$) divided by the area of a vasculature cell ($A_1$). We measured the area of vascular cells (n = 19) using ImageJ, and averaged them to determine $A_1$ (*Supplementary file 3*). Although the diffusion coefficient returned by RICS is from a population of vascular and endodermal cells, we assumed that it is a good approximation of SHR movement between one vascular and one endodermal cell. We included a second active transport term for movement in the reverse direction, from the endodermis to the vasculature, where $a_2$ is the active transport rate. However, based on the pCF analysis, $a_2 = 0$ since normally there is no bidirectional movement. Adding linear degradation gave us the equation for the change in SHR in the vasculature over time.

$$\frac{dS_v}{dt} = k_1 - a_1 S_v + a_2 S_e - d_1 S_v$$

$$a_1 = \frac{D_1}{A_1}, \; a_2 = 0$$

Given that SHR is not produced in the endodermis, there is no production term in the equation. Thus, the only SHR present in the endodermis is the SHR that moves from the vasculature. This leads to the equation for the change in SHR monomer in the endodermis over time.

$$\frac{dS_e}{dt} = a_1 S_v - a_2 S_e - d_2 S_e$$

$$a_2 = 0$$

The SHR homodimer forms from two SHR monomers. Our N&B analysis revealed that SHR homodimer does not form in a SCRi line. This suggested that the homodimer formation rate, $k_2$, should depend on the concentration of SCR. To account for this, we modeled $k_2$ as a logistic function of the concentration of SCR. Once the SCR concentration reaches a critical value $C_0$, $k_2$ will increase at rate $k$ until it reaches a maximum value of $L$ (*Figure 5—figure supplement 2*). We chose values for $C_0$, $k$, and $L$ based on the N&B data (see Parameter Estimation). Thus, the equation for the change in SHR homodimer over time is

$$\frac{dS_{2e}}{dt} = k_2(C)S_e{}^2 - d_3 S_{2e}$$

$$k_2(C) = \frac{L}{1 + e^{-k(C-C_0)}}.$$

Unlike the other variables, SCR production is not a linear term but rather a Hill equation. This structure was chosen because it has been shown that the SHR-SCR complex activates SCR expression (*Cui et al, 2007*). We assumed that both the 1:1 and 2:1 SHR-SCR complexes can activate SCR. In addition, SCR has been shown to autoregulate itself (*Cui et al, 2007*; *Moreno-Risueno et al, 2015*). Therefore, the change in SCR over time is expressed as:

$$\frac{dC}{dt} = k_3 \left( \frac{K_{1D}{}^2 C + K_{1D} SC + S_2 C}{K_{1D}{}^2 K_{2D} + K_{1D} K_{2D} S_v + K_{1D}{}^2 C + K_{1D} SC + S_2 C} \right) - d_4 C.$$

Finally, our N&B analysis revealed that both the monomer and homodimer of SHR can bind SCR and form a complex. The final two equations in our model measure the change in these complexes over time.

$$\frac{dSC}{dt} = k_4 S_e C - d_5 SC$$

$$\frac{dS_2 C}{dt} = k_5 S_{2e} C - d_6 S_2 C$$

In addition to the wild type model, we constructed a model that simulates SCRi (Model 2). Our pCF analysis revealed that SHR movement in the SCRi line is bidirectional, so now $a_2$ is defined as the experimentally determined diffusion coefficient ($D_2$) divided by the area of an endodermal cell ($A_2$). We determined the average area of endodermal cells by averaging the area of representative cells (n = 19) as we did for vascular cells (*Supplementary file 3*). Since it had been shown that a 60% reduction of SCR is required to produce the mutant layer, we assumed that SCR concentrations are maintained below 60% and that the change in SCR over time is zero (*Cui et al., 2007*).

$$\frac{dS_v}{dt} = k_1 - a_1 S_v + a_2 S_e - d_1 S_v$$

$$\frac{dS_e}{dt} = a_1 S_v - a_2 S_e - d_2 S_e$$

$$\frac{dS_{2e}}{dt} = k_2(C)S_e^2 - d_3 S_{2e}$$

$$\frac{dC}{dt} = 0$$

$$\frac{dSC}{dt} = k_4 S_e C - d_5 SC$$

$$\frac{dS_2 C}{dt} = k_5 S_{2e} C - d_6 S_2 C$$

$$a_1 = \frac{D_1}{A_1}, \; a_2 = \frac{D_2}{A_2}, \; k_2(C) = \frac{L}{1 + e^{-k(C-C_0)}}$$

## Sensitivity analysis

The sensitivity analysis was performed to determine the most influential parameters in our model. Notably, we reasoned that small changes in highly influential parameters could result in large

changes in the model outcome. In addition, it has been shown that parameter estimation can become more computationally complex and produce more uncertainty in the parameter values as the number of estimated parameters increases (*Smith, 2014*). Therefore, we focused on estimating only the most influential parameters. We chose to use Sobol decomposition to measure how sensitive the model is to a particular parameter (*Sobol, 2001*). Sobol decomposition is a variance-based method, meaning that the sensitivity of the model to a parameter is quantified by calculating the variance in the model outcome. In addition, Sobol decomposition is a global sensitivity method, so we are exploring the entire parameter space in order to determine the most influential parameters (*Smith, 2014*).

Since the Sobol decomposition allows us to calculate numerous Sobol indices, we chose to use the total effects index to measure sensitivity in our model. Accordingly, the total effects index takes into account how sensitive the model is to a single parameter as well as combinations of more than one parameter. Thus, by using the total-effects index, we take into account any parameter interactions in our model.

To calculate the Sobol total effects index, we rewrote model 1 in the form $Y = f(X_1, X_2, \ldots, X_{15})$ where the $X_i$ represent the 15 parameters. $Y$ is the model outcome, which in this case must be a scalar. The ODE solution is a set of values over time, so the solution must be numerically integrated to obtain a single, constant value (*Smith, 2014*). Consequently, the total effects index for parameter $i$, $S_{Ti}$, is defined as

$$S_{Ti} = \frac{E_{\mathbf{X}_{\sim i}}(V_{X_i}(Y|\mathbf{X}_{\sim i}))}{V(Y)} \qquad (1)$$

where $E(\,)$ denotes the expected value, $V(\,)$ denotes the variance, and $\mathbf{X}_{\sim i}$ is the vector of parameters without the $i$th parameter value. The numerator represents the expected variance in the model if all factors except $X_i$ are fixed. If this term is very large, then that means that $X_i$ contributes greatly to variance in the model: in other words, $X_i$ is an influential parameter. We divided by the variance in the model outcome so that the value of the total index is normalized across different outcomes (*Saltelli et al, 2010*; *Sobol, 2001*).

We calculated the index for each parameter using Monte Carlo evaluations and built two matrices A and B, which are 1000 rows by 15 columns. Each row represented a random draw of all 15 parameters from the parameter space (*Supplementary file 3*). For each parameter $i = 1,2,\ldots,15$, we constructed a third matrix C that was identical to A except that the $i$th column of A was replaced with the $i$th column of B. Model 1 was then evaluated for all 1000 rows. When we evaluated the ODE, we obtained 6 different outcomes: one for each variable we measured. To understand how each parameter affects all of the different variables we calculated the total effects index for each variable separately.

After evaluating the model, we numerically integrated the solution to obtain a constant value. The result is a 1000 by 6 matrix, where each row represents a model evaluation and each column represents one of the variables (SHR in the vasculature, SHR monomer, etc). We then used each column to evaluate (1), giving us the total effects index for parameter $i$ with respect to each of the variables. Since the numerator of (1) cannot be computed exactly, we estimated it using one of the most accurate estimators,

$$E_{\mathbf{X}_{\sim i}}(V_{X_i}(Y|\mathbf{X}_{\sim i})) \approx \frac{1}{2N}\sum_{j=1}^{N}\left(f(\mathbf{A})_j - f(\mathbf{C})_j\right)^2,$$

where $N$ is the number of Monte Carlo iterations (in our case, $N = 1000$) (*Saltelli et al., 2010*). Finally, we repeated the entire process 10 times to obtain technical replicates (*Supplementary file 4*). We used the Wilcoxon test with the Steel-Dwass test for multiple comparisons at significance level $\alpha = 0.1$ to determine parameters that are significantly more influential.

## Parameter estimation

For the initial model simulations, we derived most of the model parameters from the previous model of SHR and SCR in the CEI (*Supplementary file 3*). It has been shown that the mutant layer in the SCRi line occurs when the concentration of SCR drops to 60% of WT levels (*Cui et al., 2007*). Therefore, for the homodimer formation term $k_2$, we chose $C_0$ to be 60% of the steady-state value of SCR.

By choosing this value, the homodimer will not form until the concentration of SCR is above 60%. Additionally, we chose $k = 0.1$ so that homodimer formation occurs rapidly after SCR passes the threshold (*Supplementary file 3*). We expected that our model would have high levels of SHR monomer and 1:1 complex relative to the levels of SHR homodimer and 2:1 complex as shown by our N&B data. However, using these parameters, we were unable to replicate these results.

We reasoned that estimating better values for some of the parameters could improve our model simulation. Thus, we used the results from our sensitivity analysis to identify the most influential parameters to estimate. We chose to estimate $d_2$, $k_3$, $K_{2D}$, and $d_4$ based on their high sensitivity indices (*Figure 5* and *Figure 5—figure supplement 1*). Although $k_1$ and $d_1$ are influential parameters, we chose not to estimate them as varying their values only changes the overall level of SHR and not the oligomeric dynamics. We also chose to estimate $L$, which is the maximum value of $k_2$. Although $k_2$ is not as influential as some of the other parameters, its functional structure comes from the N&B analysis, so its maximum value should be derived from the N&B data.

For the parameter estimation, we started parameters at their default values (*Supplementary file 3*) and varied them one at a time until we were able to replicate the N&B data. To replicate the N&B data, we required that the SHR homodimer accounted for 7.5% of the total SHR proteins, while the 2:1 complex accounted for 15.2% of the total SHR-SCR complexes (*Figure 3* and *Figure 4*). We found that increasing $d_2$, decreasing $K_{2D}$, and decreasing $L$ achieved this relationship. In addition, decreasing $K_{2D}$ resulted in SCR increasing over time (*Figure 5D*). We did not change the values of $k_3$ and $d_4$. Using N&B to experimentally determine the oligomeric state and protein-protein stoichiometry of SHR and SCR allowed us to estimate better values for some parameters and, further, improve the reliability of the conclusions we derived from the model.

## Acknowledgements

NMC is supported by a NSF GRF (DGE-1252376). CMW is supported by a NRSA F32 (GM-106690-01). EG is supported by NIH (P41-GM103540 and P50-GM076516). APF is supported by a NSF GRF (DGE-1252376). This work was funded by a grant to PNB from the NSF Arabidopsis 2010 program (IOS-1021619), from the NIH (R01-GM043778), by the Howard Hughes Medical Institute and the Gordon and Betty Moore Foundation (through Grant GBMF3405), and by a NSF CAREER grant (MCB-1453130) to RS.

## Additional information

### Funding

| Funder | Grant reference number | Author |
| --- | --- | --- |
| National Science Foundation | MCB-1453130 | Rosangela Sozzani |
| National Science Foundation | DGE-1252376 | Natalie M Clark<br>Adam P Fisher |
| National Institutes of Health | GM-106690-01 | Cara M Winter |
| National Institutes of Health | P41-GM103540 | Enrico Gratton |
| National Institutes of Health | P50-GM076516 | Enrico Gratton |
| National Science Foundation | IOS-1021619 | Philip N Benfey |
| National Institutes of Health | R01-GM043778 | Philip N Benfey |
| Howard Hughes Medical Institute | GBMF3405 | Philip N Benfey |
| Gordon and Betty Moore Foundation | GBMF3405 | Philip N Benfey |

The funders had no role in study design, data collection and interpretation, or the decision to submit the work for publication.

## Author contributions

NMC, Designed the scanning FCS experiments, Conducted the experiments, Analyzed the data, Designed and analyze the mathematical model, Wrote the paper; EH, CMW, Analyzed the scanning FCS data, Helped revise the paper; APF, Collecting the majority of additional data the reviewers requested in their decision, specifically collected additional RICS data on SHR:SHR-GFP in vascular and QC cells and additional pCF data on SHR:SHR-GFP in shr2 and SCRi in endodermal and cortical cells; GC, Conducted the scanning FCS experiments, Analyzed the data; IB, Provided molecular material; EG, Participated in experimental design, Assisted with data analysis, Helped revise the paper; PNB, Participated in experimental design, Assisted with designing the mathematical model, Help write and revise the paper, Analysis and interpretation of data; RS, Designed the scanning FCS experiments, Conducted experiments, Analyzed data, Assisted with designing the mathematical model, Helped revise the paper

## Author ORCIDs

Natalie M Clark, http://orcid.org/0000-0003-0988-321X
Philip N Benfey, http://orcid.org/0000-0001-5302-758X

# Additional files

### Supplementary files

• Supplementary file 1. Results of shapiro-wilk goodness of fit (GoF) test on RICS and N&B data.

• Supplementary file 2. PSF beam waist values.

• Supplementary file 3. Parameter values used for mathematical modeling.

• Supplementary file 4. MATLAB code used to compute the Sobol total effects index.

### Major datasets

The following dataset was generated:

| Author(s) | Year | Dataset title | Dataset URL | Database, license, and accessibility information |
|---|---|---|---|---|
| Natalie M Clark, Elizabeth Hinde, Cara M Winter, Adam P Fisher, Giuseppe Crosti, Ikram Blilou, Enrico Gratton, Philip N Benfey, Rosangela Sozzani | 2016 | Data from: Tracking transcription factor mobility and interaction in Arabidopsis roots with fluorescence correlation spectroscopy | http://dx.doi.org/10.5061/dryad.50410 | Available at Dryad Digital Repository under a CC0 Public Domain Dedication |

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
