## [Decision Letter]

Thank you for submitting your article "Tracking transcription factor mobility and interaction in *Arabidopsis* roots with fluorescence correlation spectroscopy" for consideration by *eLife*. Your article has been reviewed by three peer reviewers, and the evaluation has been overseen by a Reviewing Editor and Detlef Weigel as the Senior Editor.

The reviewers have discussed the reviews with one another and the Reviewing Editor has drafted this decision to help you prepare a revised submission.

Summary:

To elucidate molecular mechanisms that control organismal growth and development, the authors have characterized two key transcription factors (SHORTROOT [SHR] and SCARECROW [SCR]) that control root radial patterning. Their earlier studies have revealed that SHR moves from the vasculature into the adjacent cell layer where it activates SCR expression, which is proposed to delimit further SHR movement through hetero-dimerisation. To probe the underlying mechanism, the authors have employed innovative techniques based on Fluorescent Correlation Spectroscopy (FCS) to non-invasively and quantitatively study SHR-SCR mobility and interactions. These tools allowed them to measure the diffusion of the GFP-tagged species, to determine that a binding partner slows this diffusion, to show that the presence of the binding partner in some cells creates a directional net motion of the factor towards the cell with the partner, and to make a measure of the likely stoichiometry of the factor and this partner. These conclusions are made possible by innovations in the measurement of species inside cells, often using aspects of the data that others merely ignore or discard and extends these tools to applications in the intact *Arabidopsis* root.

Essential revisions:

All three reviewers found the application of FCS techniques and the N&B analysis potentially useful tools in the arsenal of understanding multicellular development, but all felt that the techniques needed to be presented in a way more accessible to readers with improvements to both the text and figures. In addition, integrating the parameters obtained using these techniques into a mathematical model of SHR-SCR regulation would demonstrate that this is biologically significant work by providing new functional insights, which the reviewers felt was needed for publication in *eLife*.

1) Both the figures and the text need to be presented in a clearer way. The authors present techniques (that they say are novel) in a fashion more appropriate for techniques that are in everyday use. The reviewers (who are experts in these techniques) found the descriptions difficult to follow, and thus for many readers the paper will be impenetrable. A couple of sentences of introduction at the beginning of each section on what the technique is, or a cartoon in some cases would help tremendously to solve this. The authors also use things that are equivalent in their minds somewhat too interchangeably and without using the terms to help the reader understand. As an example of how the text impresses rather than illuminates, consider these sentences in the introduction: "To that end, we explored the possibility of combining scanning Fluorescence Correlation Spectroscopy (scanning FCS) techniques. These are different from the more common time correlation FCS techniques as they are based on the probability of finding a molecule at a given distance from a starting point with a time delay (Petrasek and Schwille, 2008, Digman and Gratton, 2011). In particular, we employed spatial temporal probability techniques to measure protein movement, protein-protein interactions, and the stoichiometry of protein complexes."

Tell us what you have done, clearly. Tell us what it means, clearly. If you do, then people will want to refer to this publication as a foundational assembly/application of tools.

2) Integrating the parameters obtained using these elegant biophysical techniques into a mathematical model of SHR-SCR regulation would help assess their significance and provide much greater mechanistic insight than currently provided in the manuscript. A model such as that described by co-author Cara Winter (http://grantome.com/grant/NIH/F32-GM106690-01) would be appropriate.

3) There are numerous places where figure presentation could be improved. More attention needs to be paid to details of presentation and to keeping consistent nomenclature. Specific examples have been included in the "minor points" section, but these should all be addressed carefully.

[Editors' note: further revisions were requested prior to acceptance, as described below.]

Thank you for resubmitting your work entitled "Tracking transcription factor mobility and interaction in *Arabidopsis* roots with fluorescence correlation spectroscopy" for further consideration at *eLife*. Your revised article has been favorably evaluated by the Reviewing editor and Detlef Weigel as the Senior editor.

The manuscript has been significantly improved and now is much more accessible as a resource. There is a remaining issue in Figure 5 that need to be addressed before acceptance.

The major message and conclusions of Figure 5, which represent the newly added model, is quite difficult to understand. At the very least, the four graphs should be given titles to make it clear that a-c represent the WT condition over increasing lengths of time and d is the SCR-RNAi line. With as much white space as if present the graphs, the keys could be made more informative (e.g., SHR-mono vs. SHR-HD) and the first sentence of the legend could summarize the conclusion from the models, not just that the graphs represent models.

---

## [Author Response]

*Essential revisions:*

*All three reviewers found the application of FCS techniques and the N&B analysis potentially useful tools in the arsenal of understanding multicellular development, but all felt that the techniques needed to be presented in a way more accessible to readers with improvements to both the text and figures. In addition, integrating the parameters obtained using these techniques into a mathematical model of SHR-SCR regulation would demonstrate that this is biologically significant work by providing new functional insights, which the reviewers felt was needed for publication in eLife.*

*1) Both the figures and the text need to be presented in a clearer way. The authors present techniques (that they say are novel) in a fashion more appropriate for techniques that are in everyday use. The reviewers (who are experts in these techniques) found the descriptions difficult to follow, and thus for many readers the paper will be impenetrable. A couple of sentences of introduction at the beginning of each section on what the technique is, or a cartoon in some cases would help tremendously to solve this. The authors also use things that are equivalent in their minds somewhat too interchangeably and without using the terms to help the reader understand. As an example of how the text impresses rather than illuminates, consider these sentences in the introduction: "To that end, we explored the possibility of combining scanning Fluorescence Correlation Spectroscopy (scanning FCS) techniques. These are different from the more common time correlation FCS techniques as they are based on the probability of finding a molecule at a given distance from a starting point with a time delay (Petrasek and Schwille, 2008, Digman and Gratton, 2011). In particular, we employed spatial temporal probability techniques to measure protein movement, protein-protein interactions, and the stoichiometry of protein complexes."*

*Tell us what you have done, clearly. Tell us what it means, clearly. If you do, then people will want to refer to this publication as a foundational assembly/application of tools.*

We thank the reviewers for their critical comments on our manuscript, especially on the fact that we need to present the figures and text in a clearer manner. To this end, we have added a cartoon to each of Figure 1–Figure 4 that illustrates each of the scanning FCS techniques. In addition to the schematics, we have added sentences to the beginning of each Results section that briefly describe the techniques used. We rewrote sections of the main text in order to make both the techniques and our results more accessible to the reader. We made sure that we included the rationale behind using all of the different techniques so that the reader understands when each technique is appropriate. Specifically, we now say why we use the technique, how it is performed, and what the results tell us. Although we have added more explanatory material to the main text, we made sure that the Materials and methods section still contains most of the details about each of the scanning FCS methods, as well as how to use the SimFCS software.

2) Integrating the parameters obtained using these elegant biophysical techniques into a mathematical model of SHR-SCR regulation would help assess their significance and provide much greater mechanistic insight than currently provided in the manuscript. A model such as that described by co-author Cara Winter (http://grantome.com/grant/NIH/F32-GM106690-01) would be appropriate.

The reviewers suggested integrating the parameters obtained using our techniques into a model of SHR-SCR regulation. Accordingly, we constructed an ordinary differential equation (ODE) model of SHR and SCR in the *Arabidopsis* root. Using a sensitivity analysis, we show that the diffusion coefficient of SHR is an extremely sensitive parameter, motivating the use of these biophysical techniques to experimentally determine protein movement. Further, we estimated some of the other parameter values using our oligomeric state and complex stoichiometry data from N&B and Cross N&B. Our model reveals new mechanistic insights about SHR and SCR dynamics in the root. Specifically, our model predicts that SCR expression is activated approximately 3 hours after influx of SHR. This prediction is supported by previous experiments in which SCR expression in a SHR inducible system is observed after 3 hours of induction (Sozzani et al. 2010, Figure 1 and Figure 1—figure supplement 1). Further, we show that the 2:1 complex forms later than the 1:1 complex (9 hours vs 3 hours) and that the entire system reaches a steady state between 18 and 24 hours. Our model allows us to better understand how SHR movement and SHR-SCR complex formation significantly contribute to root patterning.

*3) There are numerous places where figure presentation could be improved. More attention needs to be paid to details of presentation and to keeping consistent nomenclature. Specific examples have been included in the "minor points" section, but these should all be addressed carefully.*

All figures now have cell types clearly labeled, and the figure legends now indicate what the different colors in each figure represent. In addition, we have ensured that the nomenclature across the paper is consistent. Below, we address each of the minor points specifically.

[Editors' note: further revisions were requested prior to acceptance, as described below.]

*The manuscript has been significantly improved and now is much more accessible as a resource. There is a remaining issue in Figure 5 that need to be addressed before acceptance.*

*The major message and conclusions of Figure 5, which represent the newly added model, is quite difficult to understand. At the very least, the four graphs should be given titles to make it clear that a-c represent the WT condition over increasing lengths of time and d is the SCR-RNAi line. With as much white space as if present the graphs, the keys could be made more informative (e.g., SHR-mono vs. SHR-HD) and the first sentence of the legend could summarize the conclusion from the models, not just that the graphs represent models.*

We thank the reviewers for their comments on Figure 5. We have added titles to all four panels (a-d) of Figure 5 to make it clear that a-c represent the simulation of wild type conditions, while d represents the simulation of the SCR RNAi line. We spelled out all of the variables in the keys. In addition, for each panel we have added a sentence in the figure legend describing the conclusions so that the reader can better understand the model.